# Study on coal dust diffusion law and new pneumatic spiral spray dedusting technology at transfer point of mine cross roadway

Deji Jing[1,2,3☯], Hongwei Liu[1,2,3], Tian Zhang[1,2,3☯]*, Shaocheng Ge[4], Shuaishuai Ren[1,2,3], Mingxing Ma[1,2,3]

**1** College of Safety Science and Engineering, Liaoning Technical University, Fuxin, China, **2** Research Institute of Safety Science and Engineering, Liaoning Technical University, Fuxin, China, **3** Thermodynamic Disasters and Control of Ministry of Education, Liaoning Technical University, Fuxin, China, **4** Safety and Emergency Management Engineering College, Taiyuan University of Technology, Taiyuan, China

☯ These authors contributed equally to this work.
* zhangtian@lntu.edu.cn

**Data Availability Statement:** All relevant data are within the paper.

**Funding:** The National Natural Science Foundation of China (51704146) JDJ Natural science fund

## Abstract

In order to solve the problem of coal dust pollution at the transfer point, a three-dimensional numerical model of wind flow-coal dust at the loading point of underground rubber run was established by computational fluid dynamics (CFD) discrete particle model and finite element method and k-ε turbulence model, and the coal dust diffusion pollution phenomenon caused by the coal flow transfer under the intersection of wind flow in the cross tunnel was studied. Based on the simulation results of wind flow velocity contour, pressure contour and isochronous flow vector distribution, the influence mechanism of wind flow and coal dust characteristics on the distribution of wind flow and coal dust diffusion in the roadway is analysed, and a dust control and reduction system and treatment scheme with new pneumatic screw spray technology as the core is proposed to suppress coal dust pollution at the reloading point. The results of the study show that the wind flow distribution is mainly influenced by the intersection of tape traction and cross-roadway wind flow, showing a complex multi-layer distribution along the roadway and in the normal direction; the diffusion of coal dust of different particle sizes is influenced by the roadway wind flow, and coal dust with particle sizes in the range of 10μm~20μm is more easily diffused, and the dust with particle sizes in the range of 20μm~45μm is mainly collected and suspended near the vortex wind flow at the cross-roadway. The coal dust in the range of 20 μm~45 μm is more likely to gather in the vortex; the treatment system effectively controls the coal dust inside the dust cover, and the spiral-shaped transported droplet particle group formed by the pneumatic spiral spray combines with it efficiently, which verifies the dust control and reduction effect of the pneumatic spiral spray system at the transfer point, and the dust removal efficiency reaches 89.35% ~93.06%, which provides relevant theoretical support for the treatment of dust pollution at the coal transfer point in underground coal mines It provides the theoretical support and means to control dust pollution at underground coal transfer points.

project in Liaoning Province (2020-MS-304) JDJ
Liaoning provincial funding for scientific research
projects (LJK0323) JDJ Funders play a supporting
role in research design, data collection and
analysis, publication decisions or manuscript
preparation.

**Competing interests:** The author declares that
there are no competing interests.

# 1 Introduction

Pneumoconiosis is currently the most serious occupational disease in China accounting for
about 90% of the total number of occupational diseases, China has more than 975,000 cases in
2019, including 873,000 cases of occupational pneumoconiosis, the number of new cases of
pneumoconiosis in China from 2007 to 2019 and accounting for the total number of occupa-
tional diseases increased year by year, the number of cases of occupational pneumoconiosis in
the last decade increased from 10,963 cases to a maximum of 28,088 in 2016 cases. Coal dust is
a serious health hazard to workers, dust not only causes damage to mechanical equipment
used in mining coal mines, but also reduces the quality of the environment and the visible
range of workers, while leading to pneumoconiosis, a serious occupational disease [1–3]. In
addition to bringing occupational diseases such as pneumoconiosis to workers, dust pollution
also increases the likelihood of underground explosions [4–6]. More than 90% of existing coal
mines in China are underground, and coal dust is mainly generated from drum cutting and
propulsion during mine coal mining [7,8], and underground coal transfer and retransfer trans-
port is one of the main factors of underground coal dust pollution [9,10]. Since the 1950s,
many researchers [11,12] have systematically investigated the dust diffusion law with wind
[13,14]. For the dust production, diffusion mechanism and its characteristics at the transfer
point, the FLUENT-based dust diffusion at the transfer point of the belt conveyor was investi-
gated and the diffusion law of dust at the coal mining face was studied [15] studied and numer-
ically simulated the effect of turbulent airflow disturbance of coal chips on coal dust pollution
characteristics, airflow migration and coal dust dispersion, and established a negative pressure
secondary dust removal device [16,17]. However, as conventional spray dust control measures
were not effective, Torano et al. confirmed the feasibility of computational fluid dynamics
(CFD) for modelling dust particles in the working face tunnel to further predict airflow and
dust within the working face; a method that successfully overcame the shortcomings of tradi-
tional calculation methods [18]. CFD was used to conduct a study of dust transport patterns in
continuous working faces and to propose a novel method of dust suppression for ventilation.
A reasonable dust management technology solution can ensure safe coal mine production
while improving the sustainability of the coal mining process [19,20]. However, despite exten-
sive research on spraying and improvements in related theoretical techniques, there are still no
effective control measures at high dust concentrations at the transfer point [21–24]. There are
few studies at home and abroad that combine the transfer process with the wind flow direction
and propose control solutions. Therefore, in order to achieve the desired dust control effect, a
numerical study on the distribution of wind flow characteristics and coal dust dispersion in
coal transfer is carried out at the No. 2 transfer point of Min Dong No. 1 mine's 00 working
face.

To this end, this paper proposes a dust control and reduction system and treatment
scheme with a new pneumatic spiral spray technology as the core, and develops a device for
installing a dust cover at the intersection of the transfer point. In this method, on the one
hand, a mist curtain formed by a spiral spray is used inside the dust hood to capture coal
dust particles and suppress coal dust pollution. On the other hand, two auxiliary pneumatic
atomising nozzles provide further dust reduction and effectively reduce the dust concentra-
tion in the tunnel. This method effectively solves the problem of heavy dust contamination at
the coal transfer point. In order to determine the effectiveness of the dust hood and to guide
its further development and application, the paper uses a fluid dynamics (CFD) based dis-
crete particle model and a finite element method and k-ε turbulence model to simulate and
analyse the internal wind flow field and the trajectory of the droplet particles in the dust
hood. The nozzle was then selected and a dust control and reduction system and treatment

solution with a new pneumatic spiral spray technology at the core suitable for coal transfer points was investigated by optimising the atomisation performance of the nozzle. The field application shows that the developed dust cover can effectively prevent coal dust from spreading and improve the efficiency of suppressing coal dust pollution, which greatly improves the working environment.

## 2 Mathematical model

COMSOL Multiphysics is a multi-field coupled calculation software based on finite element theory. The basic module is quite complete in theory and can be supplemented with specialised solution modules for the joint solution of multiple physical fields. In this study, the k-ε turbulence model is firstly used to establish a three-dimensional computational mathematical model of the characteristic distribution of the wind flow field at the reproduction point. Secondly, the DEM method is used to assume the particles as discrete masses, and the fluid flow particle tracking module is used to track the released particles and obtain the results of the spatial and temporal distribution of the various characteristics of the particles at each spatial location. Finally, the directional switching of the inlet and outlet of the wind flow is used to control the relative direction of the wind flow and the coal transfer (Deji Jing et al., 2021[12]; Zhang Tian et al., 2020) [20].

CFD is used to calculate flow field data for the continuous phase of the fluid, assuming that the fluid conforms to the equations for conservation of momentum, mass and energy.

1. The conservation of momentum equation, from Newton's second law, gives

$$\frac{\partial(\rho u_x)}{\partial t} + \nabla \cdot (\rho u_x \vec{u}) = -\frac{\partial p}{\partial x} + \frac{\partial \tau_{xx}}{\partial x} + \frac{\partial \tau_{yx}}{\partial y} + \frac{\partial \tau_{zx}}{\partial z} + \rho f_x$$

$$\frac{\partial \rho \left(\rho u_y\right)}{\partial t} + \nabla \cdot \left(\rho u_y \vec{u}\right) = -\frac{\partial p}{\partial y} + \frac{\partial \tau_{xy}}{\partial x} + \frac{\partial \tau_{yy}}{\partial y} + \frac{\partial y}{\partial z} + \rho f_y \qquad (1)$$

$$\frac{\partial(\rho u_z)}{\partial t} + \nabla \cdot (\rho u_z \vec{u}) = -\frac{\partial p}{\partial x} + \frac{\partial \tau_{xz}}{\partial x} + \frac{\partial \tau_{yz}}{\partial y} + \frac{\partial \tau_{zz}}{\partial z} + \rho f_z$$

where $u_x$, $u_y$, $u_z$ velocity components, m/s; $\rho$: fluid density, kg/m$^3$; $\tau_{xx}$, $\tau_{xy}$, $\tau_{xz}$ viscous stress, N; $f_x$, $f_y$, $f_z$: force per unit mass, m/s$^2$.

2. Conservation of mass equation

$$\frac{\partial \rho}{\partial t} + \frac{\partial \rho(u_x)}{\partial x} + \frac{\partial \rho\left(u_y\right)}{\partial y} + \frac{\partial \rho(u_z)}{\partial z} = 0 \qquad (2)$$

where $u_x$, $u_y$, $u_z$ velocity components, m/s; $\rho$: fluid density, kg/m$^3$.

3. The energy equation, from the first law of thermodynamics, gives:

$$\frac{\partial(\rho E)}{\partial t} + \nabla \cdot [\vec{u}(\rho E + p)] = \nabla \cdot \left[ k_{eff} \nabla T - \sum_j h_j J_j + \left(\tau_{eff \cdot u}\right) \right] + S_h \qquad (3)$$

where $E$: total energy of the sum of internal, kinetic and potential energy, J/kg; $E = h-p/\rho + u^2$; $h$: enthalpy, J/kg; $h_j$ is the enthalpy of component $j$, J/kg; $k_{eff}$ is the effective heat transfer coefficient, W/(m·k); $k_{eff} = k + k_j$, $k$: turbulent heat transfer coefficient; $J_j$: diffusive flux of component $j$; $S_h$: volumetric heat source term.

It is assumed that the dust particles are mainly subject to gravity, buoyancy, traction and lift forces. The equations of motion for dust particle dynamics:

$$m_p = \frac{du_p}{dt} = F_g + F_f + F_d + F_x \tag{4}$$

Where: $m_p$ is the mass of the solid particle, mg; $u_p$ is the speed of movement of the solid particle, m/s; $F_d$ is the resistance to the particle, N; $F_g$ is the particle's own gravity, N; $F_f$ the buoyancy of the airflow to which the particle is subjected, N; $F_x$ is the other forces acting on the particle, including: gravity, buoyancy, stoke traction, lift, N.

The turbulence model is assumed to fit the turbulent kinetic energy equation (k equation):

$$\frac{\partial(\rho k)}{\partial t} + \frac{\partial(\rho k u_i)}{\partial x_i} = \frac{\partial}{\partial x_i}\left[\left(\mu + \frac{\mu_i}{\sigma_k}\right)\frac{\partial(k)}{\partial x_j}\right] + G_k - \rho\varepsilon \tag{5}$$

Turbulent energy dissipation rate equation ($\varepsilon$):

$$\frac{\partial(\rho\varepsilon)}{\partial t} + \frac{\partial(\rho\varepsilon u_i)}{\partial x_i} = \frac{\partial}{\partial x_i}\left[\left(\mu + \frac{\mu_i}{\sigma_\varepsilon}\right)\frac{\partial(\varepsilon)}{\partial x_j}\right] + \rho C_1 E\varepsilon - \rho C_2 \frac{\varepsilon^2}{k + \sqrt{v\varepsilon}} \tag{6}$$

Where: $C_1 = \max[0.43, \eta/(\eta+5)]$, $\eta = E_k/\varepsilon$; $C_2$ is a constant; $E = \sqrt{2E_{ij} + E_{ij}}$; $G_k$ turbulent kinetic energy generation, $G_k = \mu_i E^2$, $\mu_t$ is the viscosity coefficient; $\sigma_k$, $\sigma_\varepsilon$ respectively k equation and $\varepsilon$ equation of turbulent flow in the flow field Prandtl coefficient, the calculation to take the empirical value $C_2$ is 1.9, $\sigma_k = 1.0$, $\sigma_\varepsilon = 1.2$.

## 3 Construction of numerical modeling physical model and meshing

In the process of meshing, it should not be too thin or too rough to mesh according to the actual situation, otherwise it will affect the calculation accuracy and calculation time of the computer. Therefore, before numerical solution, the physical model should be meshed according to the actual situation and the model.

### 3.1 The physical model

In order to verify the relative wind flow transfer simulation, the coal material is transferred from the downstream channel of the 00 working face to the east belt roadway through two transfer points, and transported from the east belt to the underground coal bunker. The dust pollution in the west roadway section below the east belt transfer point is very serious (Fig 1).

### 3.2 The mesh subdivision

The spatial distribution of each major structure in this transfer point geometric model, and the geometric model established within COMSOL software according to the production system diagram, with simple geometry instead of belt conveyor turbine, control box, etc. Ignoring the pressure air pipeline, cable space volume and roadway wall roughness, assuming the same height of coal material transported on the tape surface and the traction effect on the wind flow with generalised roughness, baffle The transfer side of the tape is different from the incoming side of the tape in terms of the degree of accumulation, and the incoming side of the coal is regarded as a zero accumulation structure mainly including the upper belt conveyor, the lower belt conveyor, and the falling coal baffle. The lower belt conveyor is arranged in the east rubber transport lane, the height difference with the upper belt conveyor is 1.5 m, the lane width is 4

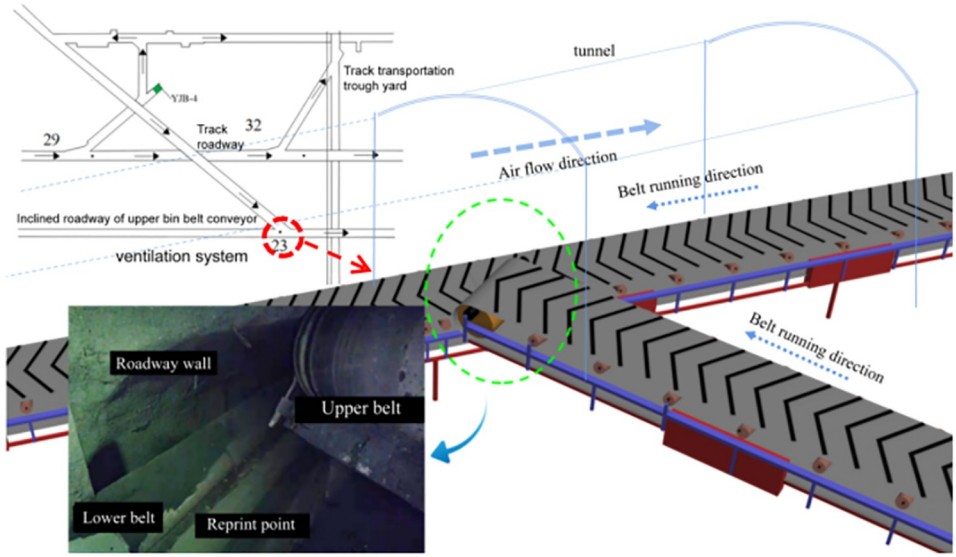

**Fig 1. Physical model of the transfer point.**

m and the height is 3.5 m. The angle between the two lanes on the horizontal plane is 39 (Fig 2).

The mesh is free-filled with a tetrahedral mesh hydrodynamic finer, with a total cell count of 6559134 and an average cell mass of 0.6752. The transfer point lane mesh size is set to 0.8, and the upper belt conveyor, lower belt conveyor and coal drop baffle are set to 0.4, with the minimum size being the default value, in order to obtain an unstructured mesh for the geometric model and make the results of the numerical simulation more accurate.

Boundary conditions are set based on experimental tests, wind speed at the lower belt conveyor transport lane is 0.2 m/s, wind speed at the upper belt conveyor contact lane is 0.15 m/s, the downhole air is relatively thin, the density is set to 1.25 kg/m$^3$, the outlet is set to free outlet, the reference pressure is 1 atm, the running speed of the belt conveyor is taken as the maximum value 4 m/s. Belt conveyor boundary settings Generalised rough wall, set at 0.8 cm according to actual coal build-up height, roughness factor 0.26. Reference temperature 293.15 K.

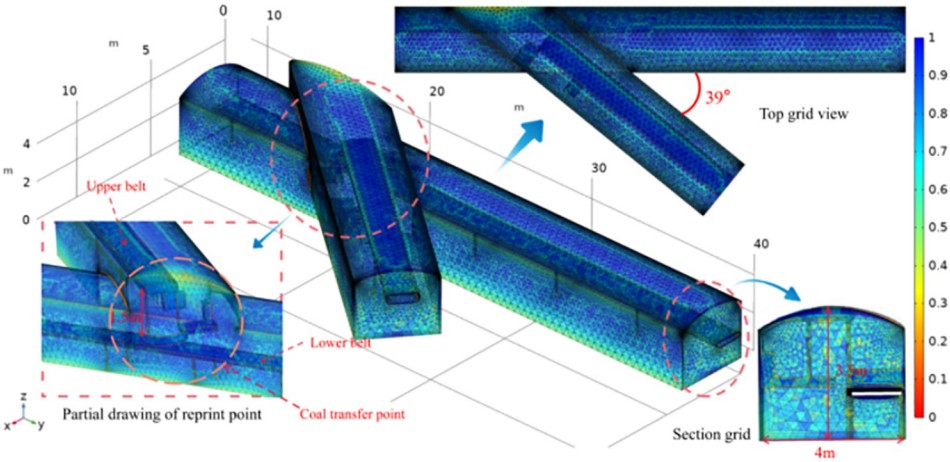

**Fig 2. Reprint point model grid diagram.**

### 3.3 Physical modeling and meshing of the device

Through the study of vortex formation at the transfer point and the ability of the spiral spray to capture and suppress the coal dust driven by the vortex, the treatment measure of setting up a spiral spray dust cover is proposed to further enclose the dust and prevent dust diffusion.

**3.3.1 Physical model of device.** As shown in Fig 3, Pneumatic spiral nozzle design based on gas-solid two-phase flow theory, confined space and free space hydrodynamic theory, aerodynamic theory. Based on the pneumatic spiral spray tank to develop a model, set the model consists of two models of the spiral nozzle and the outer field, the spiral nozzle initial setting conditions for the outer nozzle cylindrical radius of 30 mm, 50 mm high, the internal dome radius of 15 mm, the angle of 73.3˚ outside the field model set for the cylinder, the radius of 500 mm, the height of 1000 mm. The actual actual size of the device and the model ratio is 4:1, the spiral spray device is set at the center of the circle at the top of the dust cover, and according to the particle trajectory distribution of coal dust and wind flow distribution rules, two groups of spray devices are arranged in the direction of the diffusion of fine particle size dust, and the fine particles are trapped by water mist dust suppression, and the atomising nozzle is a gas-liquid two-phase pneumatic atomising nozzle, which can effectively control the disturbing wind flow at the reloading point, and The pneumatic atomising nozzle can effectively control the disturbing wind flow at the transfer point, and the micron-level droplets are well combined with dust below 20 μm, resulting in high dust reduction efficiency. The pneumatic circuit air pressure is proposed 0.4MPa, water flow rate 4 L/min. The bottom of the unit is provided with an opening (Fig 3).

**3.3.2 Device grid division.** The geometric model was meshed through the grid, and the overall quality of the grid was improved by modifying the size, shape and density of the low-quality grid. The accuracy of the numerical simulation results is closely related to the quality of the grid. Refinement is used at the transition of the section and is also used at the change in height difference angle to establish a total of 511325 grids with an average grid mass of 0.22. A

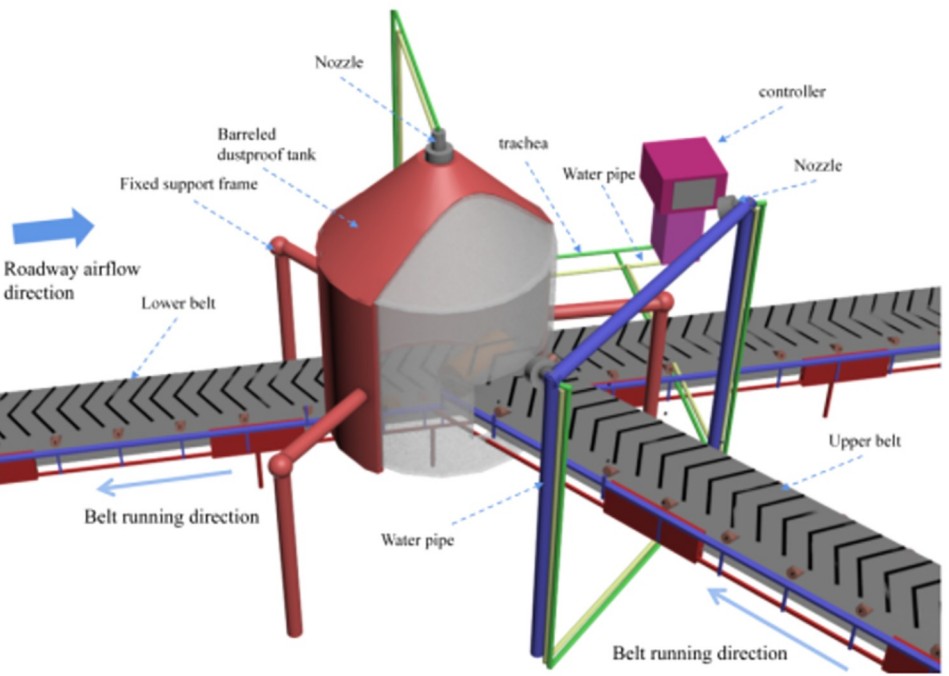

**Fig 3. Installation of pneumatic screw spray and dust removal device.**

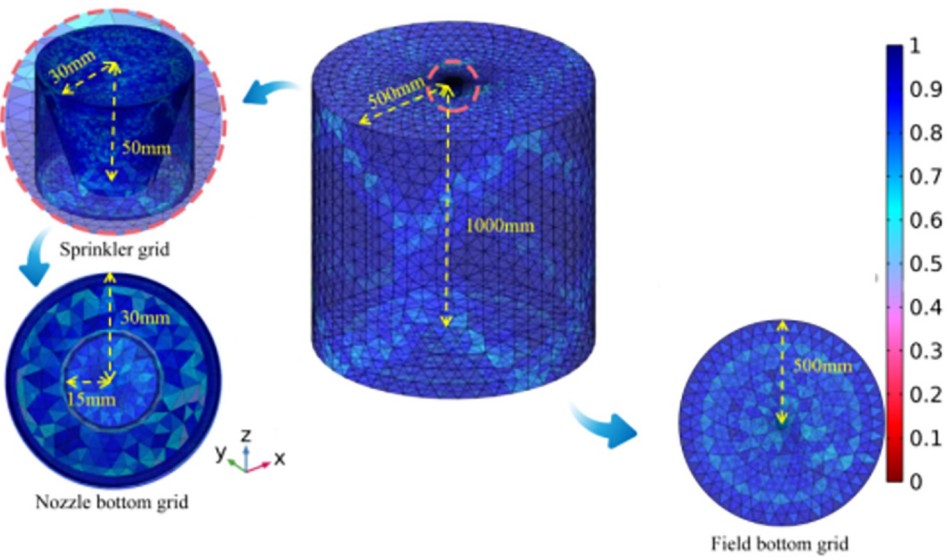

**Fig 4. Model drawing of spiral nozzle.**

dust control technology device grid is established, and the overall grid division is conducted to obtain a total of 387249 grid cells, of which the minimum cell mass is 0.005124. The boundary conditions were set based on experimental tests and simulated according to the model parameters (Fig 4). The values of the parameters set for the boundary conditions in the simulation are shown in Table 1.

## 3.4 Reliability verification and analysis

The droplet size is measured by a spray laser particle size meter, which is designed using the Fronhofer diffraction principle and a typical parallel optical path. In combination with a high performance high power laser source, the droplet testing is carried out. As shown in the histogram in Fig 5, it can be seen in the histogram that the particle size of 8μm to 15μm accounts for the largest amount when the air pressure reaches 7Mpa, and by the line graph it can be

**Table 1. Boundary conditions and parameter values.**

| Project | Name | Parameter setting |
|---|---|---|
| Boundary condition | Lower lane inlet air velocity (m/s) | 0.2 |
| | Upper lane inlet air velocity (m/s) | 0.15 |
| | Outlet pressure (pa) | 0 |
| | Air density(kg/m$^3$) | 1.25 |
| Dust source parameters | Continuous-phase dynamic viscosity (Pa·s) | 1.8×10−5 |
| | Molecular diffusion coefficient of gas (m2/s-1) | 2×10−5 |
| | Density of particles themselves (g/cm$^3$) | 1.33 |
| | Number of entry particles | 1000 |
| | Max particle size of dust (m) | 3.16e-6 |
| | Min particle size of dust (m) | 1.43e-5 |
| Droplet source parameters | Nozzle flow rate (L/min) | 11.5 |
| | Initial velocity (m/s) | 8 |
| | Half angle of atomization (°) | 40 |

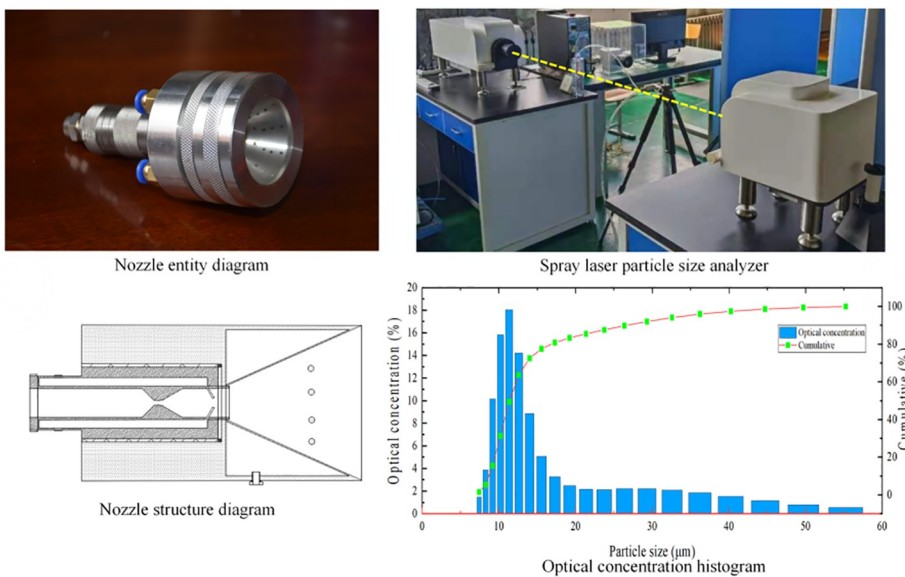

**Fig 5. Structure diagram of pneumatic spiral nozzle.**

seen that the overall particle size is below 20μm when the optical coverage has reached 80%, thus proving that the pneumatic spiral spray nozzle can achieve the required effect of dust reduction at the loading point.in the histogram can be reflected in the air pressure up to 7Mpa particle size in 8μm ~ 15μm accounted for a larger amount, the line graph shows that the particle size overall in the 20μm below the optical coverage has reached 80%, can effectively capture the transfer point diffusion of small particle size coal dust, proving that the pneumatic spiral nozzle to meet the coal transfer loading point dust reduction requirements (Fig 5).

The experimental platform was built with the following experimental apparatus: water tank, gas storage tank, pneumatic spiral nozzle, etc. This experiment takes the method of data comparison, prepare two groups of the same quality type of coal dust taken back from the site, the first group of experiments through the homemade dust collector will be sprayed into the open transparent dust box, by the coal dust collector for data collection, after one minute coal dust is still in suspension; the second group of experiments in the coal dust spray while turning on the pneumatic spiral nozzle, by the coal dust collector for data collection. The final comparison yielded a dust reduction efficiency of 93% for the spiral spray dust hood. It can be seen that the spiral airflow drives the spray to form a spiral mist curtain in a spiral shape, when the coal dust impacts the spiral mist curtain, the coal dust can be isolated and controlled in the spiral mist curtain, effectively preventing the spread of coal dust, which also coincides with the spiral advance of the spray particles in the numerical simulation, verifying the accuracy of the numerical simulation (Fig 6).

## 4 Numerical simulation analysis

### 4.1 Velocity analysis of airflow field at transfer point

The average wind speed of the whole tunnel is 0.353 m/s and the wind speed of the belt surface is 2.1 m/s. The wind speed in the lower tunnel is higher than the wind speed in the upper tunnel due to the influence of the support bar below the belt, as can be seen from the overall distribution, the wind flow velocity shows a belt distribution in both horizontal and vertical directions, longitudinally from the top plate downwards: acceleration belt along the top plate,

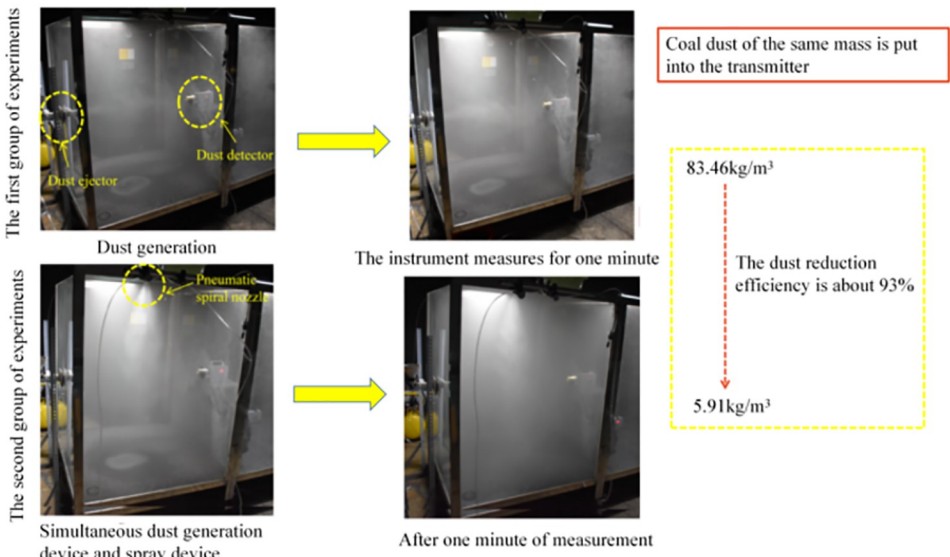

**Fig 6. Experimental results of spray dustfall.**

central buffer belt, belt surface traction acceleration belt and bottom plate deceleration belt. The horizontal direction can be divided into a near-wall acceleration belt, a mid-side deceleration mixed belt and a central belt traction reverse acceleration belt. At the front end of the transfer point where the wind converges, the wind velocity rises significantly and is influenced by the upper belt and the wind flow, the speed of the tunnel wall at the lower belt crossing reaches 0.8m/s~1.6m/s and is influenced by the baffle, the wind flow and the belt transport speed to form a vortex (Fig 7).

The four pictures (a), (b), (c) and (d) in Fig 8 are the airflow distribution of the lower roadway at different heights. Z = 1.5, Z = 2, Z = 2.5, Z = 3, respectively. The wind velocity values in

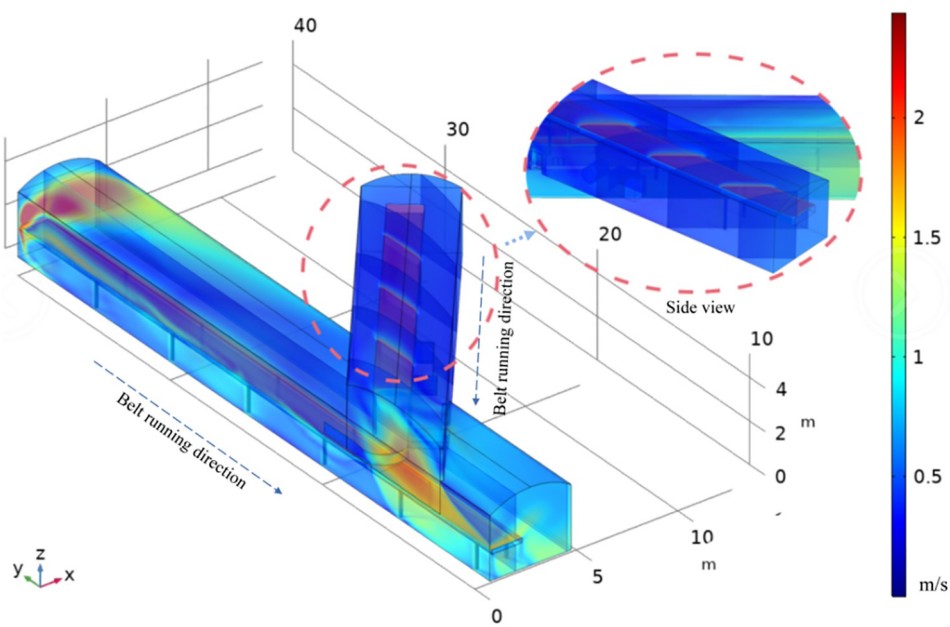

**Fig 7. Lower lane upwind coal transfer velocity distribution diagram.**

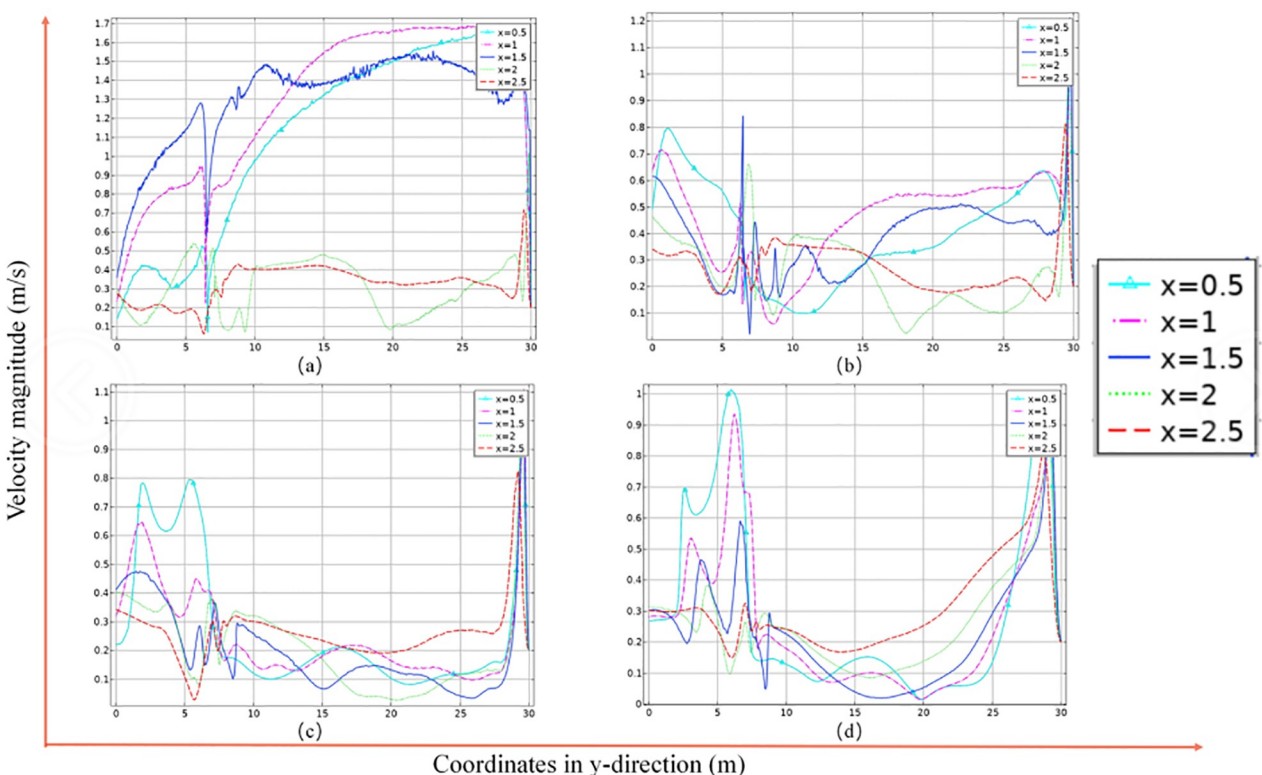

**Fig 8. Wind flow distribution curve in the centre of the lower lane.**

the xy plane on the line corresponding to different x-coordinate values, x = 0.5, x = 1, x = 1.5, x = 2, x = 2.5, are used to analyse the wind flow distribution pattern in the cross lane. At Z = 1.5 m, which is exactly where the baffle and associated structures are located, the resistance caused by them causes a large change in wind flow at Y = 6 m and mainly occurs near the belt face. The increase in velocity is evident after the wind flow crosses the baffle, which is due to the effect of the converging wind. As the simulation assumes that the air in the underground tunnel is an incompressible gas, the converging air flow from the upper tunnel increases the flow on the downwind side of the lower tunnel at a constant tunnel section, resulting in a significant increase in wind speed at a constant section. The maximum wind speed is up to 1.7 m/s and occurs at x = 0.5 m and x = 1 m. This is near the belt side of the lower tunnel, mainly due to the low resistance in this area and the inertial force of the wind flow, which is squeezed by the incoming fresh air when it hits the wall and forms an acceleration area along the wall. As shown in Fig 6, in the centre of the tunnel, as the height increases and the closer to the top of the tunnel, the horizontal wind flow velocity gradually decreases and the difference between the velocity on the downwind side and that on the incoming wind side gradually increases, this is because there are fewer structures near the top and the wind flow resistance becomes smaller. The frequency of the oscillation of the curve decreases and the amplitude decreases the closer the spatial section intercepted with the data is to the top plate at the height z = 1.5, indicating that the turbulence of the wind flow is greater below z = 2 m. Therefore, the wind flow velocity at the transfer point in three-dimensional space is obtained: the wind flow in the different zones is characterised by a greater wind speed above than below and a greater wind speed in the outlet direction than in the inlet direction. The wind flow is pulled by the roughness of the belt surface, accelerates along the belt running direction, and moves in the direction

of the tunnel wall, and gradually decays along the near-wall side of the belt in the direction of the pavement. The speed of traction wind flow on the belt surface is around 1m/s, the wind flow traction speed is affected by the baffle, the wind flow around the baffle and the belt surface traction wind flow to form a spiral vortex, near the baffle and near the wall speed in the unified range: 0.4m/s ~ 0.6 m/s. Other locations in the tunnel wind speed increases from the top plate downward, from the belt downward decay, the speed size in 0.2 m/s or less (Fig 8).

This is because the upper belt is influenced by the wind flow at the transfer point during the coal transportation process, forming a traction acceleration trend at the end, while the lower belt shows an overall deceleration trend due to the influence of the wind flow from the upper belt and the role of the baffle, and the wind flow in the tunnel is stratified, with large changes in direction and more vortices. The wind velocity in the lane is stratified, with large changes in direction and more vortices. From the above analysis, we can basically obtain the stratified distribution of wind speed, which is mainly influenced by the position of the baffle, the angle of the lane and the distance between the tape faces. Therefore, three disturbance areas are obtained, which are located at the rendezvous point, at the baffle obstruction and at the new air inlet, where the lower belt pulls the wind flow to the new air inlet during the coal transport process and forms a disturbance with the new air flow in the lane. The vector cone density is highest and the flow turbulence is strongest within about 3 m of the transfer point. The vortex formation rule can be derived from a and b: the wind flow moves through the upper and lower lane with a height difference of 1.5m and an angle of 39˚, the upper lane accelerates at the transfer point due to the traction of the upper belt surface and the wind flow velocity, the lower lane interferes with the wind flow at the head due to the traction of the lower belt floor and the velocity at the transfer point decreases. Under the influence of the angle of the lane and the height difference of the belt, the lower lane wind flow bypasses the baffle and crosses the upper lane wind flow at the front end of the transfer point at 3m, forming a vortex with an average speed of 0.45m/s (Fig 9).

## 4.2 Dust particle trajectory analysis

When the dust comes out of the transfer point for 5s, the coal dust particles transported by the upper belt fall to the lower belt through the wind baffle, during the falling process the coal dust particles are disturbed by the vortex to form a diffusion trend, part of the large diameter coal

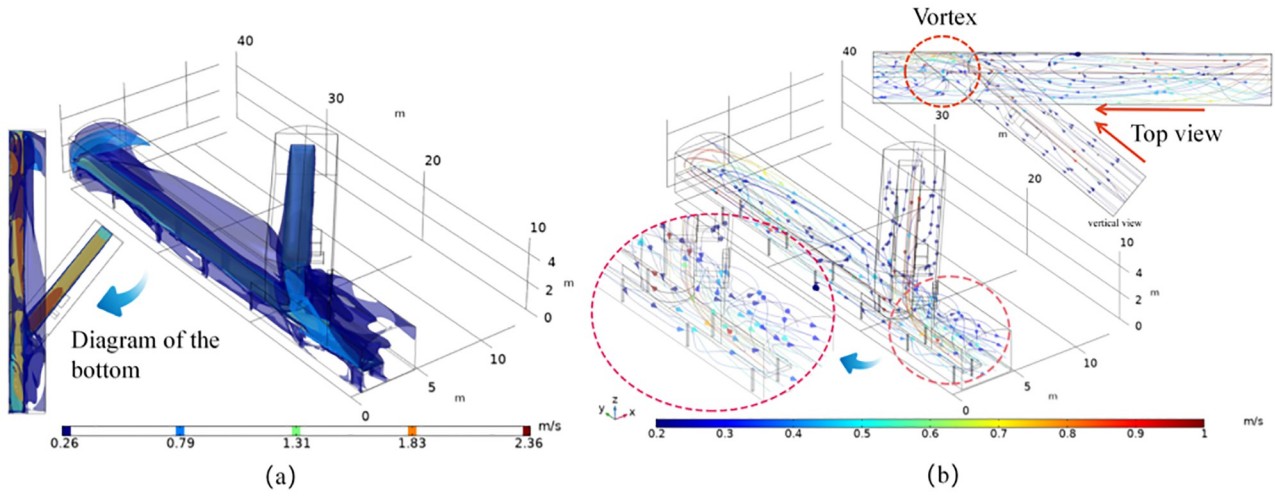

**Fig 9. Wind flow distribution characteristics and wind flow line trajectory.**

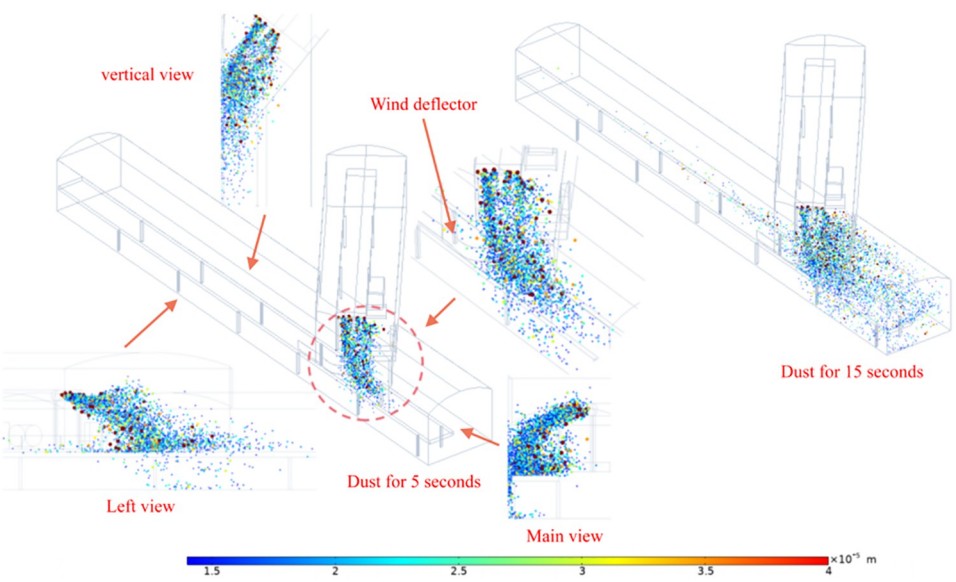

**Fig 10. Diffusion pollution range of coal dust with different particle sizes.**

dust falls to the lower belt, the remaining small diameter coal dust is dispersed with the wind flow movement, from the three views shown in Fig 10, the coal dust mainly diffuses to the downwind side, and a small part diffuses to the incoming wind side. The coal dust is mainly dispersed to the downwind side, with a small amount spreading to the incoming wind side and a portion spreading to the bottom of the belt to form a pile. At 15s after the dust has fallen, the dust is completely dispersed and spreads in a scattered state in the direction of the coal transport, and the dust with a particle size of less than 20 μm spreads over a distance of 10 m. The degree of dispersal is greatly increased, mainly due to the high induced wind speed of the dust stripped from the coal stream. The coal dust movement law can be obtained: the coal conveying direction is consistent with the wind flow direction, which leads to the coal dust spreading farther, and the smaller the coal dust particle size is, the easier it is to be pulled by the wind flow, the coal dust spreading distance is mainly affected by the belt running direction and the angle of the roadway, and the coal dust is easily spread to the downwind side by the vortex, and the coal dust concentration is large on the downwind side and small on the upwind side of the reloading point (Fig 10).

## 4.3 Analysis of wind flow field and droplet trajectory of the device

Based on the results of the wind flow distribution and particle trajectories from the tunnel simulation, a dust cover device is proposed for full vortex coverage. The nozzle internal air holes are arranged in a uniformly distributed eight-hole structure, the air holes are sprayed at the same angle, as shown in Fig 11, the airflow through the angle of the air holes and the impact of the inner wall of the nozzle to form a spiral spray, the airflow velocity at the air holes can reach 28m / s, the average airflow velocity in the dust cover can reach 2.76m / s, so through the flow line can be observed when the nozzle work to form a rotating cyclone around the cutting arm, as shown in Fig 11. The cyclone covers the entire transfer point of the dust pollution serious vortex area, and the direction of the wind flow can be observed through the direction of its rotation on the axis of the cut-off arm. The droplet particle trajectory is influenced by the airflow field to form a spiral state, the number of droplet particle size from the inside to the

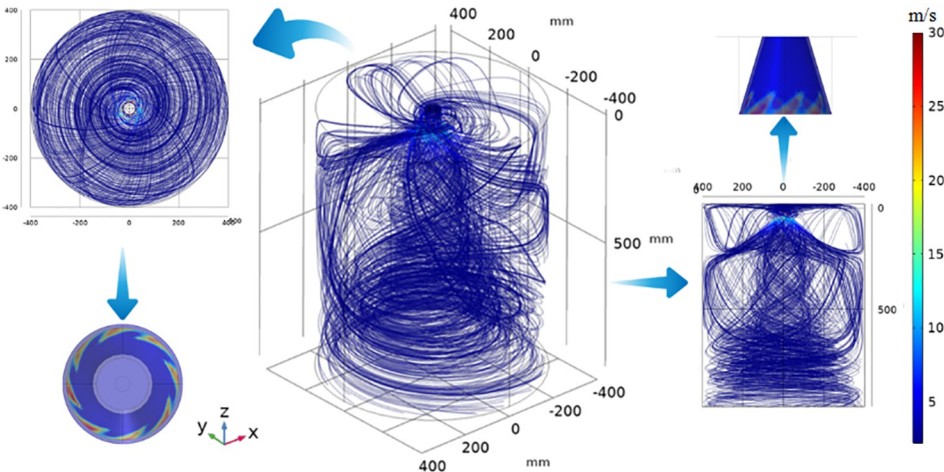

**Fig 11. Simulation of velocity field of spiral spray.**

outside to form a decreasing trend and discrete state, the smallest particle size can reach 20μm or less, it can be seen that the droplet particles cover a wide area, and the vortex area coal dust to form a counterpart, can effectively capture the dust diffusion inside the dust cover. The droplet particles produced by the spiral pneumatic spraying device are broken instantly after spraying and then impacted by the spiral high-pressure gas all around, and the particle size is less than 20μm after secondary breaking, and the droplet particles are dispersed inside the device to capture the dust with lower particle size (Figs 11 and 12).

## 5 Applications in the field

Before the application of the spiral spray dust hood to control dust, the No. 2 transfer point in the coal dust concentration, visibility is low, the amount of dust accumulation, and due to the influence of interference wind flow, so that the transfer point in the respiratory dust is difficult to settle, always in suspension free state, thus greatly increasing the chances of respiratory dust into the human body, causing human hazards. After the implementation of the treatment measures programme, the front-end 2 pneumatic atomising nozzles supplemented by the middle spiral spray dust cover when opened simultaneously, under the influence of the disturbing wind flow and vortex flow at the transfer point, can still efficiently remove and inhibit dust pollution at the transfer point, prompting a rapid decline in coal dust concentration.

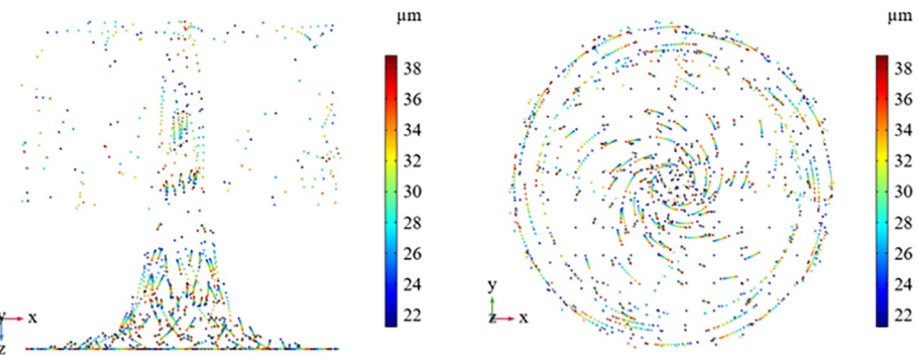

**Fig 12. Spiral pneumatic spray fog drip particle trajectory.**

**Table 2. Contrast of respiratory coal dust concentration before and after comprehensively controlling.**

| serial number | Measurement point name | Treatment, frontal coal dust concentration/(mg/m³) | Governing coal dust concentration/(mg/m³) | Dust reduction efficiency/% | Respiratory dust concentration before governance/(mg/m³) | Respiratory dust concentration after treatment/(mg/m³) | Dust reduction efficiency/% |
|---|---|---|---|---|---|---|---|
| 1 | Transfer point upper roadway upper wind side 3m | 6.84 | 4.13 | 39.62 | 2.21 | 1.98 | 10.41 |
| 2 | Dressing point lower roadway upper wind side 5m | 2.04 | 1.52 | 25.51 | 0.62 | 0.53 | 14.52 |
| 3 | Reploquette lower wind side 2M | 53.57 | 3.58 | 93.31 | 12.87 | 1.14 | 91.15 |

The numerical simulation of the transfer point is combined with the measurement of actual data in the field. On the one hand, the accuracy of the simulation results was verified through actual measurements on site. On the other hand, the numerical simulation of the pneumatic spiral nozzle effectively guided the installation location of the spray dust reduction, resulting in respirable dust and total dust control efficiencies of 91.15% and 93.31% respectively. The experimental verification of the simulation and then the proposed on-site treatment measures device and installation method effectively reduced the dust concentration at the No. 2 transfer point, as shown in Table 2, the respirable coal dust pollution concentration at each measurement point of the pneumatic spiral spray dust hood at the No. 23 transfer point was between 2.04 and 53.57 mg/m3, and the respirable coal dust pollution concentration after the section was between 0.62 and 12.87 mg/m3, and the dust removal efficiency The dust removal efficiency reached 91.15%~93.31%, which greatly improved the working environment.

## 6 Conclusion

1. The tunnel wind flow is accelerated by the belt traction and surface friction in the direction of the belt running and deflected towards the tunnel wall. The wind flow from above the belt moves around the baffle under the influence of inertia and baffle resistance through the belt traction action,with a speed between 0.4 m/s and 0.6 m/s. The wind velocity increases from the top plate downwards at other locations in the tunnel and decays from the belt downwards with velocities of 0.2 m/s or less.

2. The wind flow in the upper lane, under the influence of the upper belt surface traction and wind flow speed, reaches the reloading point for accelerated movement, the wind flow in the lower lane, under the action of the lower belt bottom plate traction wind flow, reaches the reloading point with reduced speed, and is affected by the lane angle and belt height difference, the lower lane wind flow around the baffle plate and the upper lane wind flow at the front end of the reloading point at 3m will form cross interference, resulting in The average velocity is 0.45m/s.

3. The droplet particle trajectory is influenced by the airflow field to form a spiral state, the number of droplet particle size from the inside to the outside to form a decreasing trend and discrete state, the smallest particle size can reach 20μm or less, it can be seen that the droplet particles cover a wide area, and the vortex area coal dust to form a counterpart, can effectively capture the dust diffusion inside the dust cover.

4. The method of experimentally verifying the simulation and then proposing on-site treatment measures devices and installing them effectively reduced the dust concentration at the No. 2 transfer point, the total dust mass concentration at 2m downwind of the No. 2 transfer point before treatment was 53.57 mg/m$^3$ and the respiratory dust concentration was 12.87 mg-m-3, the total dust mass concentration was 3.58 mg/m$^3$ after treatment with dust hoods, and the measured respiratory dust concentration The dust removal efficiency reached 91.15%~93.31%, which greatly improved the working environment.

## Author Contributions

**Conceptualization:** Deji Jing, Mingxing Ma.

**Data curation:** Hongwei Liu.

**Formal analysis:** Tian Zhang.

**Methodology:** Shuaishuai Ren.

**Visualization:** Shaocheng Ge.

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
