## [Decision Letter · Decision Letter 0]

27 Apr 2022

PONE-D-22-08269Study on Coal Dust Diffusion Law and New Pneumatic Spiral Spray Dedusting Technology at Transfer Point of Mine Cross RoadwayPLOS ONE

Dear Dr. zhang,

Thank you for submitting your manuscript to PLOS ONE. After careful consideration, we feel that it has merit but does not fully meet PLOS ONE’s publication criteria as it currently stands. Therefore, we invite you to submit a revised version of the manuscript that addresses the points raised during the review process.

We look forward to receiving your revised manuscript.

Kind regards,

Mohammad Mehdi Rashidi

Academic Editor

PLOS ONE

Journal Requirements:

3. PLOS requires an ORCID iD for the corresponding author in Editorial Manager on papers submitted after December 6th, 2016. Please ensure that you have an ORCID iD and that it is validated in Editorial Manager. To do this, go to ‘Update my Information’ (in the upper left-hand corner of the main menu), and click on the Fetch/Validate link next to the ORCID field. This will take you to the ORCID site and allow you to create a new iD or authenticate a pre-existing iD in Editorial Manager. Please see the following video for instructions on linking an ORCID iD to your Editorial Manager account: https://www.youtube.com/watch?v=_xcclfuvtxQ.

Reviewers' comments:

Reviewer's Responses to Questions

**Comments to the Author**

1. Is the manuscript technically sound, and do the data support the conclusions?

Reviewer #1: Yes

Reviewer #2: Yes

2. Has the statistical analysis been performed appropriately and rigorously? 

Reviewer #1: Yes

Reviewer #2: Yes

3. Have the authors made all data underlying the findings in their manuscript fully available?

Reviewer #1: Yes

Reviewer #2: Yes

4. Is the manuscript presented in an intelligible fashion and written in standard English?

Reviewer #1: Yes

Reviewer #2: Yes

5. Review Comments to the Author

Reviewer #1: The subject of the research is the study of dust control device equipment proposed for the problem of dust pollution at the transfer point, which is based on the theory of a spiral spray device and is investigated using a finite element - dynamic mesh modelling method. The reliability of the simulation is adequately given in terms of the airflow flow and dust dispersion at the transfer point site. The solution of treating the transfer point with a dust cover is proposed, the wind flow field and particle trajectory at the simulated transfer point are studied to obtain the vortex distribution law, the dust cover is installed for the vortex location, and the internal wind flow and droplet distribution are simulated. The idea is relatively new, well argued and the data is reliable. I believe that the results of the study can provide an important reference for research on dust management at mine transfer points. However, the quality of the writing of the paper still needs to be improved, and I support the publication of this paper if these following issues are addressed (with minor modifications).

1. The inconsistency of proper names in Figure 1 suggests a correction.

2. Some references are not cited in the article and need to be added.

3. Some of the images have blurred text resulting in the information in the images not being read by the reader and affecting the overall nature of the article. For example, Figure 8, Figure 10.

4. The text has a large number of images applied to the wind flow field at the point of reproduction, it is recommended that the images be combined or selectively deleted.

5. Boundary conditions are very important for numerical simulations and different boundary conditions can affect the final results of the simulation, therefore boundary conditions should be specified. It is recommended to add dust source conditions and droplet atomisation conditions.

6. The authors' figure 12 reflects the spiral spray velocity field wind flow curve and is missing units. The dust hood is also a closed device and when spraying for a longer period of time there should be a build-up of droplets and wind flow at the bottom, whether the author's device has an opening should be clarified.

7. In the design of the treatment solution in paper 4.1, the authors demonstrate that the spray device is effective in capturing small particle size coal dust by measuring the droplet size with a spray laser particle size meter, the basis of which needs to be detailed.

8. Authors should use superscripts for numbers in m³ in their conclusions.

9. The description of the experimental section is too brief and does not indicate a clear method of experimentation.

Reviewer #2: In order to solve the problem of coal dust pollution at the conveying point, the manuscript adopts computational fluid dynamics (CFD) discrete particle model, finite element method and K- ε Based on the turbulence model, the three-dimensional air flow coal dust numerical model of the loading point of the underground rubber runway is established, and the coal dust diffusion pollution caused by the coal flow movement at the air flow intersection in the cross tunnel is studied. The influence mechanism of airflow and coal dust, the distribution of airflow and the characteristics of coal dust diffusion in the roadway are analyzed. A dust control and reduction system and a treatment scheme are put forward, which are based on the new pneumatic screw spray technology, so as to suppress the coal dust pollution at the refueling point. The field application shows that the developed dust cover can effectively prevent the diffusion of coal dust and improve the efficiency of coal dust pollution control, and has a good application prospect. The manuscript is rich in content and is worth revising and publishing. However, the manuscript still has the following problems.

1. Please compare the two groups of experiments in section 3.3.4 with charts.

2. Why is the wind speed at the lower part of the tunnel greater than that at the upper part?

3. What are the characteristics of wind speed distribution in different zonal areas in horizontal and vertical directions?

4. Put forward reasonable suggestions according to the numerical simulation results.

5. It is suggested that the findings of this study be discussed more in the summary.

6. There are insufficient references, so more references need to be supplemented. The background and mechanism of seepage are not introduced clearly. In particular, The failure and damage characteristics of rock should be further described. The author should introduce this mechanism.

Ecological risk assessment of soil and water loss by thermal enhanced methane recovery: Numerical study using two-phase flow simulation. Journal of Cleaner Production, 2022, 334, 130183.

Coupled thermo-hydro-mechanical modelling for geothermal doublet system with 3D fractal fracture. Applied Thermal Engineering, 2022, 200, 117716.

6. PLOS authors have the option to publish the peer review history of their article (what does this mean?). If published, this will include your full peer review and any attached files.

Reviewer #1: **Yes: **Jianwei Cheng

Reviewer #2: No

---

## [Author Response · Author response to Decision Letter 0]

30 Apr 2022

I've put all the replies you requested into the file and submitted it. If it's not done correctly, please let me know by the editor and I'll change it again, thank you.

---

## [Decision Letter · Decision Letter 1]

30 May 2022

PONE-D-22-08269R1Study on Coal Dust Diffusion Law and New Pneumatic Spiral Spray Dedusting Technology at Transfer Point of Mine Cross RoadwayPLOS ONE

Dear Dr. zhang,

Thank you for submitting your manuscript to PLOS ONE. After careful consideration, we feel that it has merit but does not fully meet PLOS ONE’s publication criteria as it currently stands. Therefore, we invite you to submit a revised version of the manuscript that addresses the points raised during the review process.

We look forward to receiving your revised manuscript.

Kind regards,

Mohammad Mehdi Rashidi

Academic Editor

PLOS ONE

Journal Requirements:

Reviewers' comments:

Reviewer's Responses to Questions

**Comments to the Author**

1. If the authors have adequately addressed your comments raised in a previous round of review and you feel that this manuscript is now acceptable for publication, you may indicate that here to bypass the “Comments to the Author” section, enter your conflict of interest statement in the “Confidential to Editor” section, and submit your "Accept" recommendation.

Reviewer #3: All comments have been addressed

Reviewer #4: All comments have been addressed

2. Is the manuscript technically sound, and do the data support the conclusions?

Reviewer #3: Yes

Reviewer #4: Yes

3. Has the statistical analysis been performed appropriately and rigorously? 

Reviewer #3: Yes

Reviewer #4: Yes

4. Have the authors made all data underlying the findings in their manuscript fully available?

Reviewer #3: Yes

Reviewer #4: Yes

5. Is the manuscript presented in an intelligible fashion and written in standard English?

Reviewer #3: Yes

Reviewer #4: Yes

6. Review Comments to the Author

Reviewer #3: The paper has been well revised and has high guiding value for the site. It is recommended to accept the paper.

Reviewer #4: The subject of this paper is about the control of dust pollution at the coal transfer point of the mine, which is studied by the finite element-dynamic mesh modeling method, and the wind flow field and dust particle trajectory at the transfer point site are simulated by the simulation COMSOL software, and the reliability of the simulation is verified through experiments and other perspectives. The wind flow field and particle trajectory of the simulated coal transfer site are studied to obtain the vortex distribution law, and the dust cover device is proposed to control the law. The idea is very novel, the arguments are more solid and the data are reliable. I think the research results can provide valuable reference for the research of dust pollution control at the coal transfer point of mine. There is no problem with the macroscopic nature of the overall article, there are some problems that need to be revised, and I recommend publishing this article if the following problems are solved.

1. The specific location of the coal transfer point should be shown in the roadway distribution system map of picture 1.

2. 3.3.1 The experimental part of the device is recommended to be put to the later reliability verification.

3. 3.3.2 The content lacks the specific description of the device dimensions.

4. The content of 3.3.3 does not match with the described picture and needs to be revised.

5. The font in Figure 11 is different from the font of other pictures, and it is suggested to be modified.

6. 4.3 The content analysis is less, and it is suggested to add specificity.

7. Conclusion 1 suggests specific description. And there is no serial number.

7. PLOS authors have the option to publish the peer review history of their article (what does this mean?). If published, this will include your full peer review and any attached files.

Reviewer #3: No

Reviewer #4: No

---

## [Author Response · Author response to Decision Letter 1]

10 Jun 2022

Dear Editor,

I thank the editor for allowing me to revise the article to make him better. 

Yours sincerely,

Tianzhang

Reviewer #4: The subject of this paper is about the control of dust pollution at the coal transfer point of the mine, which is studied by the finite element-dynamic mesh modeling method, and the wind flow field and dust particle trajectory at the transfer point site are simulated by the simulation COMSOL software, and the reliability of the simulation is verified through experiments and other perspectives. The wind flow field and particle trajectory of the simulated coal transfer site are studied to obtain the vortex distribution law, and the dust cover device is proposed to control the law. The idea is very novel, the arguments are more solid and the data are reliable. I think the research results can provide valuable reference for the research of dust pollution control at the coal transfer point of mine. There is no problem with the macroscopic nature of the overall article, there are some problems that need to be revised, and I recommend publishing this article if the following problems are solved.

1. The specific location of the coal transfer point should be shown in the roadway distribution system map of picture 1.

Answer:Modification completed

2. 3.3.1 The experimental part of the device is recommended to be put to the later reliability verification.

Answer:Modified and added to the current paragraph 3.4, combined with the original 3.3.4 into one paragraph

3. 3.3.2 The content lacks the specific description of the device dimensions.

Answer:Modified, in what is now paragraph 3.3.1

4. The content of 3.3.3 does not match with the described picture and needs to be revised.

Answer:Modified, in what is now paragraph 3.3.2

5. The font in Figure 11 is different from the font of other pictures, and it is suggested to be modified.

Answer:Modification completed

6. 4.3 The content analysis is less, and it is suggested to add specificity.

Answer:Modification completed

7. Conclusion 1 suggests specific description. And there is no serial number.

Answer:Modification completed

---

## [Decision Letter · Decision Letter 2]

18 Jul 2022

Study on Coal Dust Diffusion Law and New Pneumatic Spiral Spray Dedusting Technology at Transfer Point of Mine Cross Roadway

PONE-D-22-08269R2

Dear Dr. zhang,

We’re pleased to inform you that your manuscript has been judged scientifically suitable for publication and will be formally accepted for publication once it meets all outstanding technical requirements.

Kind regards,

Mohammad Mehdi Rashidi

Academic Editor

PLOS ONE

Additional Editor Comments (optional):

Reviewers' comments:

Reviewer's Responses to Questions

**Comments to the Author**

1. If the authors have adequately addressed your comments raised in a previous round of review and you feel that this manuscript is now acceptable for publication, you may indicate that here to bypass the “Comments to the Author” section, enter your conflict of interest statement in the “Confidential to Editor” section, and submit your "Accept" recommendation.

Reviewer #3: All comments have been addressed

2. Is the manuscript technically sound, and do the data support the conclusions?

Reviewer #3: Yes

3. Has the statistical analysis been performed appropriately and rigorously? 

Reviewer #3: Yes

4. Have the authors made all data underlying the findings in their manuscript fully available?

Reviewer #3: Yes

5. Is the manuscript presented in an intelligible fashion and written in standard English?

Reviewer #3: Yes

6. Review Comments to the Author

Reviewer #3: The paper has been fully revised to meet the requirements of the journal and is recommended to be published.

7. PLOS authors have the option to publish the peer review history of their article (what does this mean?). If published, this will include your full peer review and any attached files.

Reviewer #3: No

---

## [Editor Report · Acceptance letter]

28 Jul 2022

PONE-D-22-08269R2 

Study on Coal Dust Diffusion Law and New Pneumatic Spiral Spray Dedusting Technology at Transfer Point of Mine Cross Roadway 

Dear Dr. Zhang:

I'm pleased to inform you that your manuscript has been deemed suitable for publication in PLOS ONE. Congratulations! Your manuscript is now with our production department. 

Kind regards, 

on behalf of

Professor Mohammad Mehdi Rashidi 

Academic Editor

PLOS ONE